# A Highly Sensitive and Flexible Strain Sensor Based on Dopamine-Modified Electrospun Styrene-Ethylene-Butylene-Styrene Block Copolymer Yarns and Multi Walled Carbon Nanotubes

**DOI:** 10.3390/polym14153030

**Published:** 2022-07-26

**Authors:** Bangze Zhou, Chenchen Li, Zhanxu Liu, Xiaofeng Zhang, Qi Li, Haotian He, Yanfen Zhou, Liang Jiang

**Affiliations:** 1College of Textiles and Clothing, Qingdao University, Qingdao 266071, China; bangzezhou@outlook.com (B.Z.); chenchenli3@outlook.com (C.L.); zhanxu_liu@outlook.com (Z.L.); xiaofengzhang0824@163.com (X.Z.); liqi980908@163.com (Q.L.); hht99tj@163.com (H.H.); 2College of Textiles, Donghua University, Shanghai 200051, China

**Keywords:** SEBS, MWCNTs, yarn, dopamine, strain sensor

## Abstract

As wearable electronic devices have become commonplace in daily life, great advances in wearable strain sensors occurred in various fields including healthcare, robotics, virtual reality and other sectors. In this work, a highly stretchable and sensitive strain sensor based on electrospun styrene-ethylene-butene-styrene copolymer (SEBS) yarn modified by dopamine (DA) and coated with multi-walled carbon nanotubes (MWCNTs) was reported. Due to the process of twisting, a strain senor stretched to a strain of 1095.8% while exhibiting a tensile strength was 20.03 MPa. The strain sensor obtained a gauge factor (GF of 1.13 × 10^5^) at a maximum strain of 215%. Concurrently, it also possessed good stability, repeatability and durability under different strain ranges, stretching speeds and 15,000 stretching-releasing cycles. Additionally, the strain sensor exhibited robust washing fastness under an ultrasonic time of 120 min at 240 W and 50 Hz. Furthermore, it had a superior sensing performance in monitoring joint motions of the human body. The high sensitivity and motion sensing performance presented here demonstrate that PDA@SEBS/MWNCTs yarn has great potential to be used as components of wearable devices.

## 1. Introduction

Strain sensors have the ability to convert physical deformations into measurable electrical signals. As wearable electronic devices have become commonplace in daily life, great advances in strain sensors occurred in various fields including healthcare, robotics, virtual reality and other sectors [1]. However, conventional strain sensors fabricated by using metals and semiconductor materials, despite their toughness, have low sensitivity and work over small strain ranges, so do not meet the requirements of high strain and high flexibility [2]. To resolve these deficiencies, an elastic polymer has been used as a substrate to provide higher mechanical deformation for strain sensors. Recently, thermoplastic polyurethane (TPU) [3,4], styrene-butadiene-styrene copolymer (SBS) [5,6], polydimethylsiloxane (PDMS) [7,8], Ecoflex [9], styrene-ethylene-butene-styrene copolymer (SEBS) [10,11] amongst others have been extensively employed in the fabrication of flexible strain sensors. Recently, resistance-based strain sensors have gradually attracted more attention because of their simple manufacturing process, low energy consumption in operation and relatively simple reading system [12,13]. At the same time, conductive nanomaterials are an important constituent in the preparation of resistance-based strain sensors. Carbon-based nanomaterials including carbon nanotubes (CNTs) [14,15], carbon blacks (CBs), graphene [2,7,16], and graphene oxide (GO) [17] are good candidates, because of their excellent electrical and robust mechanical properties [18,19]. 

At present, various methods have been applied to prepare strain sensors, but most sensors cannot simultaneously possess both high workable strains and high sensitivity [20]. Chen et al. [16] chose CBs as the conductive material and rubber as the substrate material to fabricate conductive composites, and then modified the composite material with PDMS to fabricate a flexible strain sensor. The gauge factor (GF, which is used for the characterization of sensitivity) reached 242.6 under a 71.4% strain. Wang et al. [21] reported a stretchable strain sensor fabricated by coating single-wall carbon nanotubes (SWCNTs) into an elastic cotton/polyurethane (PU) core-spun yarn. It had a maximum workable strain up to 200%, but the GF was only 0.06. Lee et al. [22] fabricated a strain sensor by incorporating a cracked transparent epitaxial layer of indium tin oxide (ITO) on a transparent PET substrate, which had a maximum GF of 4000 while its sensing strain was only 2%. Consequently, the fabrication of a strain sensor with both a high strain capability and a high GF presented a challenge, which is described in this study.

SEBS is a kind of thermoplastic elastomer with high elongations and elastic recovery properties [23,24,25,26]. However, due to its low surface energy and poor compatibility with inorganic materials and weak adhesion, SEBS requires modification before use. Dopamine (DA) has strong adhesion because it can form a polydopamine (PDA) layer via self-polymerization. PDA is similar to mucin, which is the most abundant macromolecule in mucus and secreted by mussels, exhibiting strong adhesion [27]. 

In this work, carboxylic multi-walled carbon nanotubes (MWCNTs), as the conductive fillers, were coated onto dopamine-modified SEBS yarn. Thus, a simple and practical method for fabricating PDA@SEBS/MWCNTs yarn-based strain sensor is reported. Firstly, SEBS yarn was prepared by electrospinning, rolling and twisting. Secondly, the surface of SEBS yarn was modified by dopamine. Thirdly, MWCNTs were coated onto dopamine-modified SEBS yarn by brushing and ultrasonication. Fourthly, the chemical composition, mechanical properties, sensing performance, durability and washing fastness were studied. Finally, the sensing performance of SEBS@PDA/MWCNTs yarn was demonstrated by monitoring various human joint motions (such as fingers, elbow, knuckle and wrist bending).

## 2. Experimental Procedures

### 2.1. Preparation of Yarn-Based Strain Sensors

The procedure for preparing the yarn-based strain sensor shown in Figure 1 includes three steps: (1) electrospinning the SEBS yarn; (2) DA modification; (3) MWCNTs coating.

#### 2.1.1. Preparation of Electrospun SEBS Yarn

SEBS (weight ratio of EP/PS = 67/33, Kraton, Houston, TX, USA) and Tetrahydrofuran (THF, Sinopharm, Beijing, China) were mixed with a SEBS concentration of 25 wt.% for 12 h at room temperature. The spinning solution was loaded into a 20 mL syringe with a 22G needle and fixed on the syringe pump. The whole electrospinning process was conducted at room temperature, while the injection speed was 3 mL/h, the voltage was 10 kV, and the receiving distance was 15 cm (Figure 1a). After 5 min, the SEBS fibrous membrane was dried at 70 °C for 1 h. Then the electrospun SEBS fibrous membrane was rolled and twisted (Figure 1b), forming the fabricated electrospun SEBS yarn.

#### 2.1.2. Dopamine Modification of the Electrospun SEBS Yarn

DA solution with a concentration of 10 mmol/L was prepared by dissolving dopamine hydrochloride (DA·HCl, Shanghai Macklin, Shanghai, China) powder into a tris(hydroxymethyl)methyl aminomethane (Tris, Beijing Solarbio, Beijing, China)-hydrochloric acid (HCl, Sinopharm, Beijing, China) solution (pH = 8.5). Then sodium periodate (SP, Shanghai Macklin, Shanghai, China) was added into the DA solution with a DA to SP mole ratio of 3:2. Finally, SEBS yarn was immersed in the DA solution for 6 h (Figure 1c), then washed with distilled water and dried in the oven at 70 °C. The mechanism of dopamine polymerization can be seen in Figure 1e. The DA-modified SEBS yarn obtained was denoted as PDA@SEBS.

#### 2.1.3. Preparation of MWCNTs Coated Composite Yarn

MWCNTs (Shenzhen Tuling, Shenzhen, China) were added to absolute alcohol (EtOH, Sinopharm, Beijing, China) to prepare MWCNTs/EtOH suspensions with different concentrations. As shown in Figure 1d, the MWCNTs/EtOH suspension was coated on the yarn using a brush. Subsequently, the MWCNTs coated yarn was soaked into suspension for ultrasonic treatment at 240 W and 50 Hz for a range of times before drying at 70 °C. The composite yarn was denoted as PDA@SEBS/MWCNTs.

**Figure 1 polymers-14-03030-f001:**
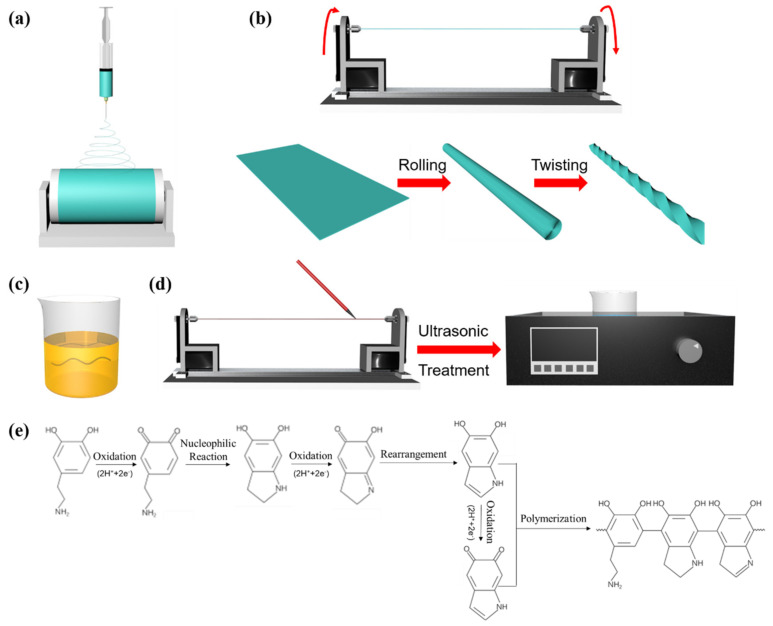
Illustrations of fabrication of the yarn-based strain sensors: (**a**) Electrospinning the SEBS fibrous membrane. (**b**) The electrospun SEBS yarn twisting process. (**c**) The DA modification of the SEBS yarn. (**d**) MWCNTs coated on PDA@SEBS yarn. (**e**) The mechanism of dopamine polymerization.

### 2.2. Characterization

The morphology of the yarns was observed by using scanning electron microscopy (SEM, TESCAN VEGA3, TESCAN, Brno, Czech Republic) with different magnifications and at an accelerated voltage of 10 kV and an electron beam intensity of 10 A/cm^2^. 

An energy-dispersive X-ray spectroscope (EDX, EDAX), equipped with SEM, and X-ray photoelectron spectroscopy (XPS, Axis Supra^+^, Kratos, Kawasaki, Japan) were used to analyze the elemental composition of the yarn surface.

Thermogravimetric analysis was conducted by a DSC/TG synchronous thermal analyzer (STA449 F3 Jupiter, NETZSCH, Bavaria, Germany) in a nitrogen atmosphere. 

Mechanical properties were measured by a universal tensile testing machine (Instron 5965, Illinois Tool Works Inc., Glenview, IL, USA) at a feed rate of 100 mm/min, while the distance between gauge points was 20 mm.

The electrical properties of the composite yarn were characterized by using a digital multimeter (KEYSIGHTB2901A, Keysight Technology, Santa Rosa, CA, USA).

## 3. Results and Discussion

Figure 2 shows SEM images of the SEBS yarn and the PDA@SEBS yarn and an EDS spectrum of the PDA@SEBS yarn. As can be seen from Figure 2a,b, the surfaces of the SEBS and PDA@SEBS yarns were smooth. However, the EDS spectrum and mapping images in Figure 2c–g show there were N and O elements on the surface of the PDA@SEBS yarn, which indicated that SEBS yarn was modified by DA successfully.

TGA was applied to quantify the content of MWCNTs coated on the surface of the untwisted PDA@SEBS yarn (PDA@u-SEBS) and twisted PDA@SEBS yarn (PDA@t-SEBS). As shown in Figure 3, the weight percentage of the materials decreased with the increase in temperature. When the temperature was up to 800 °C, the weight percentages of PDA@SEBS, PDA@u-SEBS/MWCNTs and PDA@t-SEBS/MWCNTs were 3.89%, 11.2% and 13.94%, respectively. The content of MWCNTs can be calculated by using Equation (1):(1)$$w_{MWCNTs}=\frac{w_{a}-w_{b}}{1-w_{b}}\times 100\%$$
where *W_a_* is the residual mass of PDA@SEBS/MWCNTs yarn and *W_b_* is the residual mass of PDA@SEBS yarn. By calculation, the contents of MWCNTs coated onto the PDA@u-SEBS and PDA@t-SEBS were 7.61% and 10.46%, respectively. This was because after twisting, the material surface was rougher, which caused a higher interfacial bond strength [28]. Thus, the twisted SEBS yarn was chosen as the matrix material of the strain sensor.

To fabricate the composite yarn so that it possessed good conductivity, the preparation process of PDA@SEBS/MWCNTs yarn was explored by a single factor experiment.

The conductivity of composited yarns coated with MWCNTs/EtOH suspension of different concentrations and 0.5-hour ultrasonic treatment can be seen in Figure 4a. With the increase in the concentration of suspension, the content of MWCNTs coated on the yarn increased (Figure 4b–h), which led to the conductivity of the composite yarn increasing. When the concentration of the MWCNTs/EtOH suspension was 12 g/L, the conductivity was 0.036 S/cm, while the conductivity was 0.037 S/cm when the concentration reached 14 g/L. Interestingly, between a concentration of 12 g/L and 14 g/L, there was almost no change in the conductivity of the composite yarns. This can be explained by employing the percolation theory [29,30]. The conductivity of the material is related to the critical concentration of conductive filler in polymer composites. The critical concentration of the conductive filler is called the percolation threshold. In the percolation threshold region, the continuous conductive network is formed only through the arrangement of fillers in conducive substrate. Above the percolation threshold, the conductivity enhanced slightly and stabilized thereafter. Accordingly, an MWCNTs/EtOH suspension concentration of 12 g/L was selected.

Figure 4i shows the conductivity of a composite yarn coated with MWCNT/EtOH suspension with a concentration of 10 g/L and ultrasonic treatment at varied times. With the increase in ultrasonic treatment time, more MWCNTs were deposited (Figure 4j–o), which was due to the cavitation effect caused by ultrasound, resulting in cavitation bubbles in the dispersion. When bubbles collapsed, powerful energy was provided to conductive particles to impact the matrix materials causing a firm adhesion to the matrix materials. The longer the time, the more conductive particles adhere [31]. However, when the ultrasonic treatment time exceeded 2.5 h, the conductivity of the composite yarn increased only slightly. Based on this consideration, the composite yarn coated with 12 g/L MWNCTs/EtOH suspension and sonicated for 2.5 h, was chosen for the strain sensor.

Figure 5 shows the XPS spectra of SEBS, PDA@SEBS and PDA@SEBS/MWCNTs yarn. It can be seen from Figure 5a that there were spectral peaks at 284.8 eV and 532 eV, representing the binding energy of C_1s_ and O_1s_, respectively, while O_1s_ peak on the curve of pure SEBS was caused by air pollution. Compared with pure SEBS, a new peak at 399 eV for N_1s_ appeared in the spectra of PDA@SEBS and PDA@SEBS/MWCNTs, indicating that the DA successfully modified SEBS yarn.

In order to further analyze the chemical composition of the materials, the C_1s_ was fitted. As shown in Figure 5b, the C_1s_ peak can be decomposed into C-C (284.8 eV), π-π* satellite peak (291.5 eV) and C-O (286 eV), while the appearance of C-O was due to air pollution. After DA modification, C-N, C=N, and C-O peaks appeared at 285.5 eV, 287.5 eV and 289 eV, respectively (Figure 5c). After deposition of MWCNTs, a new peak, C=C, appeared at 284 eV (Figure 5d). It proved that PDA and MWCNTs were coated onto the SEBS yarn successfully.

The mechanical properties are of great significance for the practical application of materials [32]. Figure 6a shows the stress-strain curves of SEBS, PDA@SEBS, and PDA@SEBS/MWCNTs yarns, the corresponding tensile strength and elongation at break are presented in Table 1. The tensile strength of SEBS yarn was 17.1 MPa. With the successive deposition of PDA and MWCNTs, the tensile strength increased, while PDA@SEBS had a tensile strength of 18.17 MPa and the value for PDA@SEBS/MWCNTs was 20.03 MPa. It was primarily because PDA act as a glue-like adhesive to firmly bond the yarn and MWCNTs. The strong interface interactions between MWCNTs and yarn act effectively to transfer stress from the yarn to the MWCNTs, so improving tensile strength. However, the elongation at break of pure SEBS (ε_at break_ = 1270.96%), compares favorably with those of PDA@SEBS and PDA@SEBS/MWCNTs yarns (1158.4% and 1095.8%, respectively). This was probably due to local stress concentrations caused by the aggregation of PDA and MWCNTs which would initiate cracks and lead to early fracture [33].

To investigate the relation between viscoelastic behavior and material repeatability, cyclic tensile tests were conducted at a tensile speed of 100 mm/min under applied strains of 5%, 10%, 50%, 100%, 200% and 300%. As can be determined from Figure 6b–d, the stress-strain for loading and unloading cycles are not coincident. There is markedly more hysteresis in the PDA@SEBS/MWCNTs yarn curves when compared with those of SEBS yarn and PDA@SEBS yarn, this is unsurprising since the extensibility of the PDA@SEBS/MWCNTs yarn was far greater than that of the two other materials. The PDA@SEBS yarn exhibited only slightly more hysteresis than the SEBS yarn. Internal friction in a material subjected to a tensile loading/unloading cycle will cause strain to lag behind stress as the polymeric chains need to achieve equilibrium as mechanical energy is converted to heat energy. So, hysteresis loops of different sizes and shapes were formed, and the area of the hysteresis loops represented the dissipated work characterized as mechanical hysteresis [34]. Useful future research can be undertaken to study the influence of hysteresis on the sensitivity of PDA@SEBS/MWCNTs devices and if the material suffers from ‘set’. Figure 6e,f show the dissipated energy of SEBS, PDA@SEBS, PDA@SEBS/MWCNTs yarns at different strains in 5 stretching-releasing cycles. With the increase in strain and the deposition of PDA and MWCNTs, the dissipated energy also increased. In addition, the dissipated energy showed the highest value in the first cycle, and then gradually decreased in the subsequent cycles until it tended to a constant value. This phenomenon is called the Mullins Effect [35]. This phenomenon was due to the irreversible deformation caused by the slip between PDA, MWCNTs and SEBS fibers and between SEBS chain segments, resulting in mechanical hysteresis. Therefore, in the first cycle, greater stress was required to reach the required elongation, while in subsequent and particularly the later cycles, only a lower force was required to reach the required elongation [2,34].

To determine the strain sensing performance, samples of 40 mm length were held in the grips, where the gauge lengths were 20 mm and copper sheets were used as electrodes. The change of resistance is characterized by relative resistance, which is given by Equation (2).
(2)$$\frac{\Delta R}{R_{0}}=\left(\frac{R-R_{0}}{R_{0}}\right)\times 100\%$$
where *R* is the real-time resistance, *R*_0_ is the initial resistance of the strain sensor. It can be seen from Figure 7a that the relative resistance increased exponentially with the increase in strain. This may be because, with an increase in strain, the MWCNTs were separated from one another, which led to the change in the number of conductive paths and the change of conductive tunnel distances between conductive fillers [30]. Further, it shows that PDA@SEBS/MWCNTs yarn had a sensing strain range of 0~215%. To study the strain sensing mechanism, the change of relative resistance of a composite yarn with a change in strain was fitted by Tunnel Effect [2,36]. The change of resistance of the composite yarn can be calculated by using Equation (3).
(3)$$R=\left(\frac{L}{D}\right)\left(\frac{8\pi hs}{3\gamma a^{2}e^{2}}\right)exp\left(\gamma s\right)$$
where *D* is the number of conductive paths and *L* is the number of particles forming a single path, *h* and *e* are Planck constant and quantum of electricity, respectively, *s* is the minimum distance between MWCNTs, and *a^2^* is the cross-section area. *γ* is a parameter representing the height of the barrier (*φ*) and a function of the electron mass (*m*), which can be calculated by Equation (4).
(4)$$\gamma =\frac{4\pi }{h}\sqrt{2\varphi m}$$

The dependance of the number of conductive paths (*N*) on the strain (*ε*) under stress is shown in Equation (5).
(5)$$N=\frac{N_{0}}{exp\left(A_{1}\varepsilon +B\varepsilon^{2}+C\varepsilon^{3}+D\varepsilon^{4}\right)}$$
where *N*_0_ is the number of conductive paths without stretch; *b*, *A*_1_, *B*, *C* and *D* are constants. Assuming that the initial distance (*s*_0_) change to *s* when suffering stretch, *s* can be obtained by Equation (6).
(6)$$s=s_{0}\left(1+b\varepsilon \right)$$

The relationship between the resistance change (Δ*R*) and strain is given by Equation (7).
(7)$$\frac{\Delta R}{R_{0}}=\frac{R-R_{0}}{R_{0}}=\left(\frac{Ns}{N_{0}s_{0}}\right)exp\left[\gamma \left(s-s_{0}\right)\right]-1$$

By substituting Equations (5) and (6) into Equation (7), the relationship between relative resistance and strain is represented by Equation (8).
(8)$$\frac{\Delta R}{R_{0}}=\left(1+\frac{b\varepsilon }{100}\right)exp(\frac{A\varepsilon }{100}+B{\left(\frac{\varepsilon }{100}\right)}^{2}+C{\left(\frac{\varepsilon }{100}\right)}^{3}\;+D{\left(\frac{\varepsilon }{100}\right)}^{4})\times 100$$
where the units of relative resistance and strain are both %, *A*= *A_1_ + γs*.

The relative resistance of composite fibers was calculated using Equation (8). As shown in Figure 7a, R^2^ is 0.99752, meaning the experimental results were very consistent with the theoretical results, in which the fitting constants *b*, *A*, *B*, *C* and *D* are 949.77242, −2.49077, 5.99487, −4.22125 and 1.01419, respectively.

The GF was given by Equation (9).
*GF* = *d*(Δ*R*/*R*_0_)/*dε*(9)
where Δ*R* = *R* − *R*_0_, *R* is the real-time resistance, and *R*_0_ is the initial resistance of the strain sensor. Figure 7b shows the GF of PDA@SEBS/MWCNTs yarn. When the strain reached a maximum of 215%, the GF was 1.13 × 10^5^. As shown in Figure 7c, compared with maximum GFs and sensing strains recently reported [2,3,6,9,10,14,15,16,21,22,37], the strain sensor in this study offered excellent, comprehensive sensing performance.

To measure the response time, composite yarn was stretched at a stretching speed of 500 mm/min and strained from 0 to 50%. The time to achieve this strain was 1.2 s. However, as shown in Figure 7d, the change of relative resistance needed 1.23 s, so the response time of the PDA@SEBS/MWCNTs yarn was 30 ms. 

To investigate the reliability and stability of the PDA@SEBS/MWCNTs yarn, the dynamic strain sensing behavior at different strain ranges and stretching speeds was determined as shown in Figure 7e,f. As can be observed from Figure 7e the PDA@SEBS/MWCNTs yarn exhibited excellent repeatability and stability in the strain range of 0~5%, 0~10%, 0~50% and 0~100% at a stretching speed of 10 mm/min. Figure 7f shows the synchronous change of resistive resistance under stretching speeds of 5, 10, 50, 100 and 200 mm/min when the PDA@SEBS/MWCNTs was strained from 0 to 50%. There was no obvious difference in the change of relative resistance. These outcomes revealed the material’s ability to detect different external stimuli.

Figure 7g shows the current (I) and voltage (V) curves of composite yarn at different strains. The I-V curves at various strains show a linear relationship, indicating compliance with Ohm’s Law. Concurrently, the long-term sensing behavior of the composite yarn was evaluated. In Figure 7h, the relative resistance of the composite yarn changed when strained from 0 to 50% and at a frequency of 0.25 Hz for 15,000 cycles. In the initial cycles, the relative resistance increased before stabilizing. This can be attributed to the continuous destruction and reconstruction of MWCNTs pathways, and then the conductive pathways stabilized at the commencement of strain cycles [38,39,40,41,42,43]. The change of relative resistance exhibited a good response in 15,000 tensile cycles, indicating PDA@SEBS/MWCNTs yarn had good resilience.

**Figure 7 polymers-14-03030-f007:**
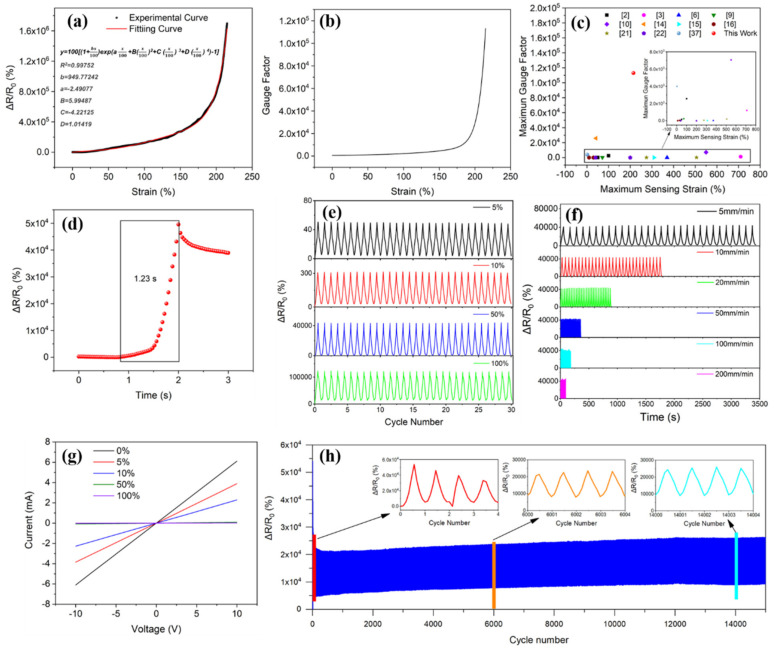
(**a**) The relative resistance change–strain curve of PDA@SEBS/MWCNTs yarn at a stretching speed of 10 mm/min. (**b**) The gauge factor–strain curve of PDA@SEBS/MWCNTs yarn at a stretching speed of 10 mm/min. (**c**) Comparison of the maximum GF and the maximum sensing strain reported in the literature in recent years and those in this work. (**d**) The response time of PDA@SEBS/MWCNTs yarn. (**e**) The relative resistance change of PDA@SEBS/MWCNTs yarn under different strains of 5%, 10%, 50%, and 100% at a stretching speed 10 mm/min. (**f**) The relative resistance change of PDA@SEBS/MWCNTs yarn under different stretching speeds of 5, 10, 50, 100 and 200 mm/min at a strain of 50%. (**g**) Current and voltage of PDA@SEBS/MWCNTs yarn under strains of 0%, 5%, 10%, 50%, and 100%. (**h**) 15,000 cycles in the strain range of 0 to 50% at a frequency of 0.25 Hz.

To characterize the washing fastness, PDA@SEBS/MWCNTs yarn was soaked in distilled water and ultrasonicated in an ultrasonic cleaner for 20, 40, 60, 80, 100 and 120 min, respectively. Then the conductivity of the composite yarn after different washing times was determined. As shown in Figure 8, the original conductivity of the composite yarn was 0.122 S/cm. Thereafter, the conductivity decreased with increases in the time of ultrasonication. After 40-min ultrasonication, the conductivity decreased to 0.16 S/cm. However, when the ultrasonic time exceeded 40 min, the conductivity changed little. After 120 min, the conductivity was 0.01 S/cm, a change rate of 91.85%. This was primarily due to the PDA layer firmly adhering MWCNTs to the SEBS matrix, so consequently the PDA@SEBS/MWCNTs had excellent water washing fastness.

As mentioned previously, PDA@SEBS/MWCNTs yarn has high flexibility, extensibility, and durability. This means that the composite yarn had potential applications in flexible wearable strain sensors. The composite yarn developed was employed to monitor human motions. As shown in Figure 9a, finger motion under different bending angles was detected when the composite yarn was attached to a consenting volunteer’s finger. Moreover, the composite yarn showed the ability to detect other joints (wrist, elbow and knee) motions under different bending angles. As can be seen from Figure 9b–d, in all cases the relative resistance increased with the increase in joint bending angle and decreased with the return of the joint to its original position.

## 4. Conclusions

In this study, a highly stretchable and sensitive strain sensor based on electrospun SEBS yarn modified by DA and coated with MWCNTs was reported. The strain sensor exhibited a high GF (1.13 × 10^5^) under a maximum strain of 215% while exhibiting good stability, repeatability and durability under a wide range of applied strains, at different stretching speeds and under long-term cyclic loading (15,000 cycles). Additionally, the strain sensor exhibited reliable washing fastness over an ultrasonic testing time of 120 min. Furthermore, it showed an enhanced sensing performance in monitoring joint motions. Although further development may lead to a lowering of hysteresis and avoidance of set when the material is used in dynamic applications, these results demonstrated that PDA@SEBS/MWNCTs yarn has great potential to be used in the field of wearable devices.

## Figures and Tables

**Figure 2 polymers-14-03030-f002:**
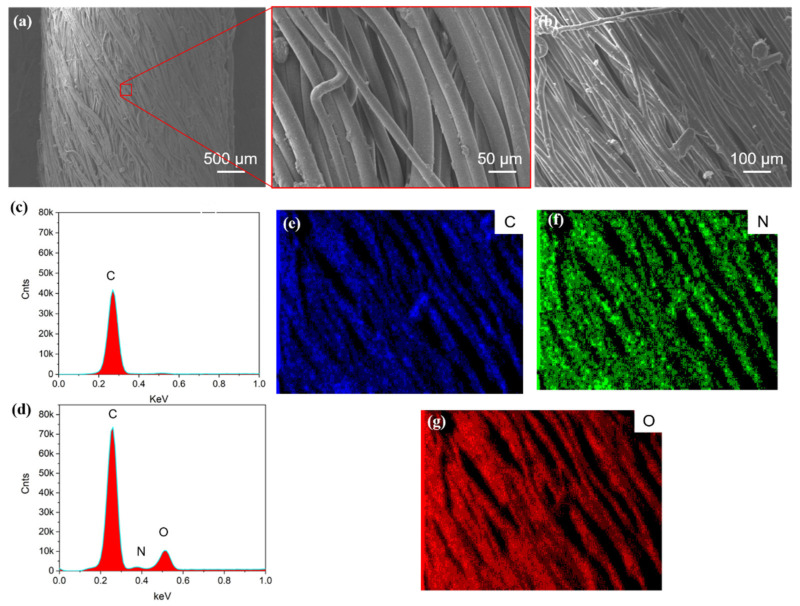
SEM images of (**a**) pure SEBS yarn and (**b**) PDA@SEBS yarn; EDX spectroscopy of (**c**) SEBS and (**d**) PDA@SEBS yarn; EDX mapping of SEBS@PDA yarn for (**e**) C, (**f**) N, and (**g**) O.

**Figure 3 polymers-14-03030-f003:**
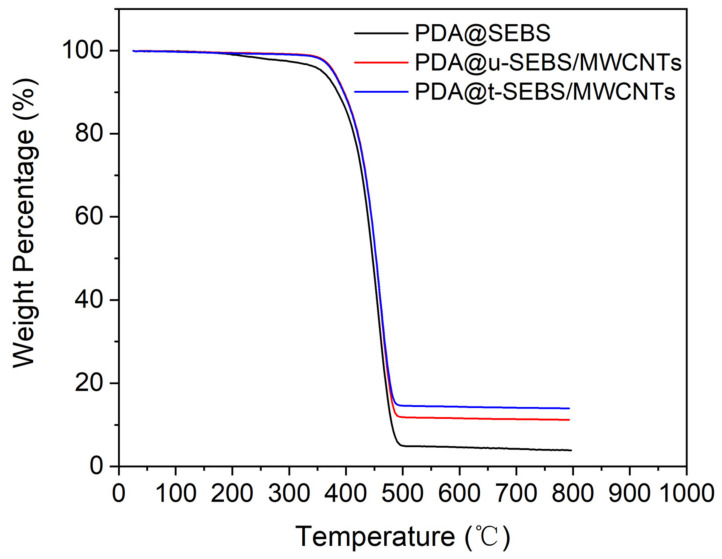
TGA curves of PDA@SEBS, PDA@u-SEBS/MWCNTs and PDA@t-SEBS/MWCNTs.

**Figure 4 polymers-14-03030-f004:**
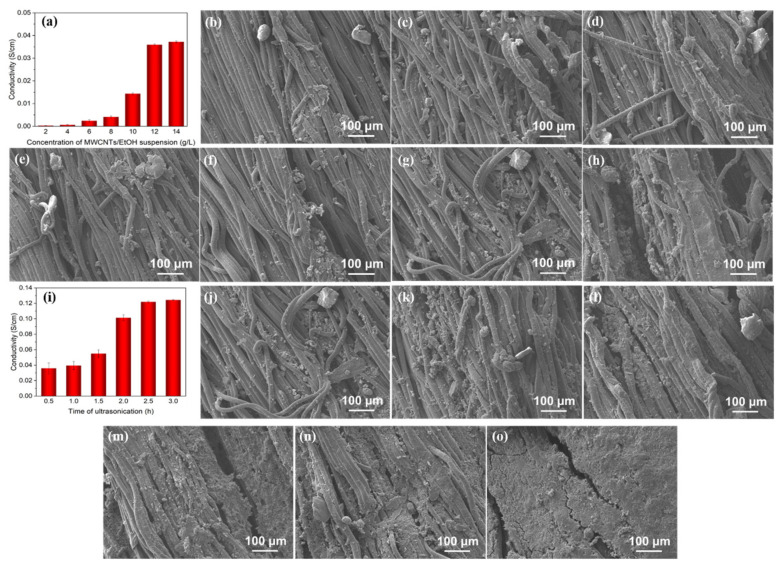
(**a**) Conductivity of PDA@SEBS/MWCNTs yarns; (**b**–**h**) SEM images of PDA@SEBS/MWCNTs yarn coated with MWCNTs/EtOH suspension concentrations of (**b**) 2 g/L, (**c**) 4 g/L, (**d**) 6 g/L, (**e**) 8 g/L, (**f**) 10 g/L, (**g**) 12 g/L and (**h**) 14 g/L; (**i**) Conductivity of PDA@SEBS/MWCNTs yarn; (**j**–**o**) SEM images of PDA@SEBS/MWCNTs sonicated for (**j**) 0.5 h, (**k**) 1 h, (**l**) 1.5 h, (**m**) 2 h, (**n**) 2.5 h, and (**o**) 3 h.

**Figure 5 polymers-14-03030-f005:**
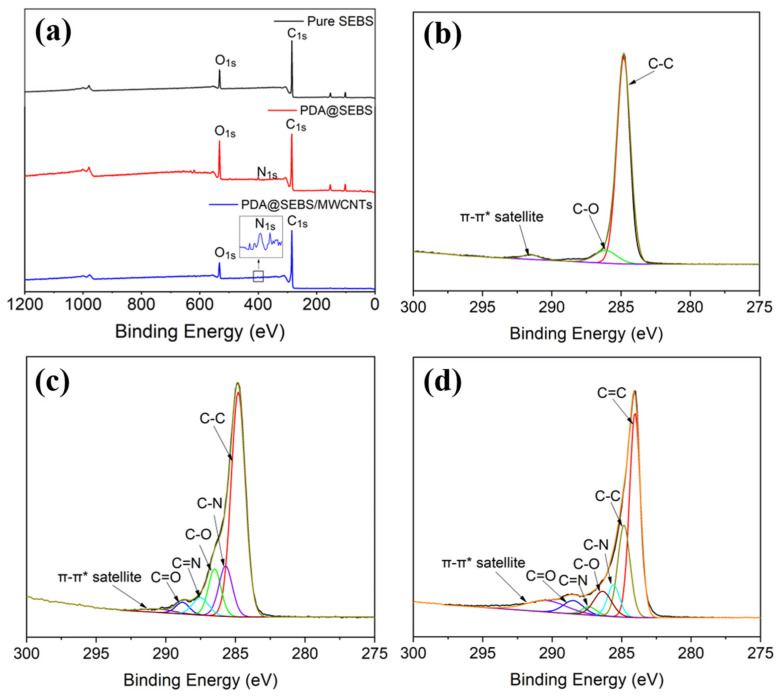
(**a**) XPS wide scan spectra of pure SEBS, PDA@SEBS and PDA@SEBS/MWCNTs. C_1s_ core-level spectra of (**b**) pure SEBS, (**c**) PDA@SEBS, (**d**) PDA@SEBS/MWCNTs.

**Figure 6 polymers-14-03030-f006:**
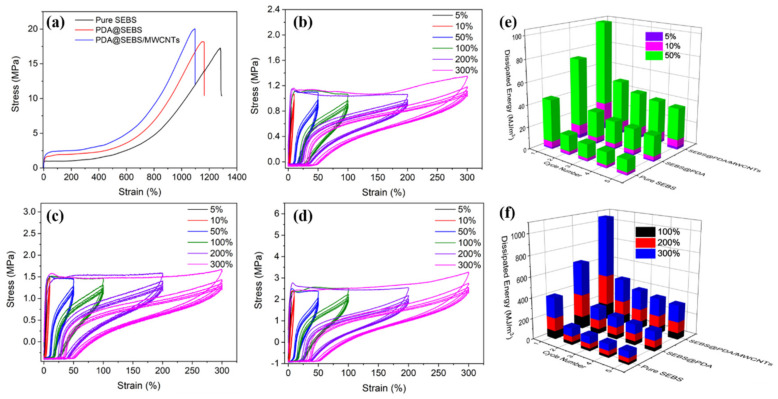
(**a**) Stress vs. strain curves of SEBS, PDA@SEBS, and PDA@SEBS/MWCNTs yarns at a feed rate of 100 mm/min. Stress vs. strain curves of the initial five tensile cycles for (**b**) SEBS yarn, (**c**) PDA@SEBS yarn, and (**d**) PDA@SEBS/MWCNTs yarn all at different strains. Dissipated energy of SEBS, SEBS@PDA and SEBS@PDA/MWCNTs yarns at the strain of (**e**) 5%, 10% and 50%, (**f**) 100%, 200% and 300%.

**Figure 8 polymers-14-03030-f008:**
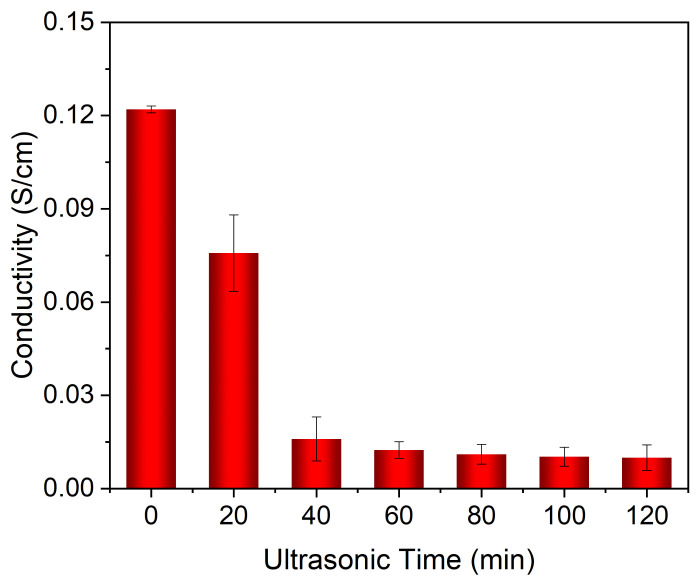
The conductivity of PDA@SEBS/MWCNTs yarn after different washing time.

**Figure 9 polymers-14-03030-f009:**
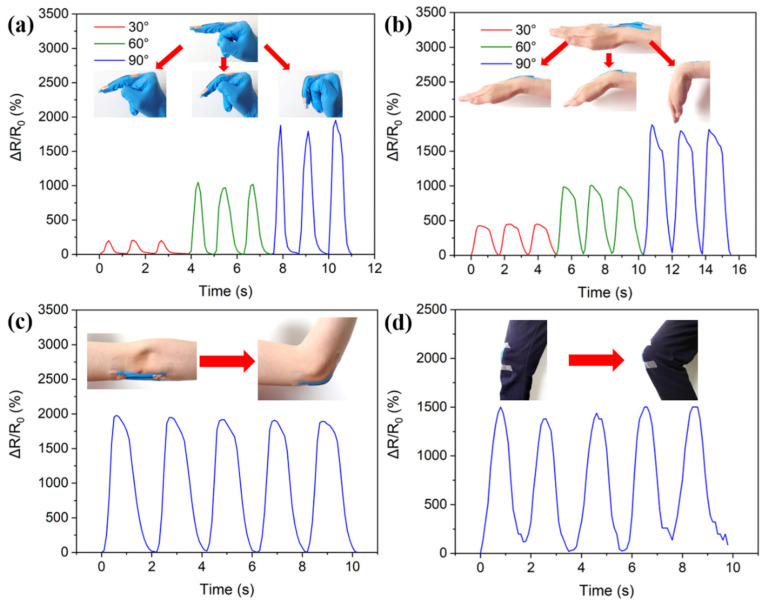
PDA@SEBS/MWCNTs yarn used to monitor human motions. (**a**) finger bending. (**b**) wrist bending. (**c**) elbow bending. (**d**) knee bending.

**Table 1 polymers-14-03030-t001:** Tensile strength (σ) and elongation at break (ε_at break_) for PDA@SEBS and PDA@SEBS/MWCNTs yarns.

Yarn	σ (MPa)	ε_at break_ (%)
SEBS	17.1 ± 0.45	1270.96 ± 100.15
PDA@SEBS	18.17 ± 0.42	1158.4 ± 97.75
PDA@SEBS/MWCNTs	20.03 ± 0.51	1095.8 ± 91.43

## Data Availability

Not applicable.

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
