# Peer review of "A Highly Sensitive and Flexible Strain Sensor Based on Dopamine-Modified Electrospun Styrene-Ethylene-Butylene-Styrene Block Copolymer Yarns and Multi Walled Carbon Nanotubes"

_polymers, 2022, doi:10.3390/polym14153030_

Round 1
Reviewer 1 Report
Review report
Manuscript title: A highly sensitive and flexible strain sensor based on dopa-mine-modified electrospun styrene-ethylene-butylene-styrene block copolymer yarns and multi walled carbon nanotubes
Type of manuscript: Article
Manuscript ID: polymers-1819832
Review comments: The work is interesting. The manuscript describes about a highly sensitive and flexible strain sensor based on electrospun SEBS yarn modified by DA and coated with MWCNTs. It is very good for the readers to this journal. But it needs the following corrections before accepting its final publication.
1. In the abstract re-written the whole part using one tense. Mixing is not correct.
2. In the abstract- As wearable electronic devices have become commonplace in everyday life,-more better- As wearable electronic devices have become commonplace in daily life.
3. In section 2.1. Preparation of yarn based strain sensors needs to short explain of preparing 4 steps individually. In the text 3 steps explain but needs serially explanations.
4. In Fig. 2 (100µm) makes it no b and rearrange the fig 2 again.
5. In Fig. 4 a & i should be inside the box of the figure like b.
6. In Fig. 5, the numbering a,b,c &d should go to the inside.
7. Fig. 6 & 7 should be the same as Fig. 5.
8. The English is also need to edit.
Author Response
The authors are most grateful for the insightful comments and the helpful recommendations suggested by the reviewers. Once addressed in the final manuscript they will be of undoubted benefit and will improve the paper accordingly. We are thankful for the opportunity to carry out the changes which are highlighted in red text in the revised manuscript.
Each point raised by the reviewers is addressed in italics text in the authors’ response following each point made by the reviewer.
Reviewer #1:
The authors agree with the reviewer and are happy to undertake the changes they suggest.
Question 1: In the abstract re-written the whole part using one tense. Mixing is not correct.
Response 1: The authors thank the reviewer for pointing this out. The tense has been revised and unified on Page 2.
Question 2: In the abstract- As wearable electronic devices have become commonplace in everyday life,-more better- As wearable electronic devices have become commonplace in daily life.
Response 2: Thank you for the useful suggestion provided by the reviewer. It has been revised on Page 2
Question 3: In section 2.1. Preparation of yarn based strain sensors needs to short explain of preparing 4 steps individually. In the text 3 steps explain but needs serially explanations.
Response 3: The authors thank the reviewer for pointing this out. The preparing steps have been corrected to 3 steps on Page 5. The process of rolling and twisting was too short to be a separate paragraph.
Question 4: In Fig. 2 (100µm) makes it no b and rearrange the fig 2 again.
Response 4: We thank the reviewer for this suggestion. In Fig. 2, there is a Figure 2b, maybe it was black and not clear, the color of “(b)” has been changed to white on Page 8.
Question 5: In Fig. 4 a & i should be inside the box of the figure like b.
Response 5: The authors thank the reviewer for pointing this out. The corresponding change has been made.
Question 6: In Fig. 5, the numbering a,b,c &d should go to the inside.
Response 6: The authors are grateful for the reviewer pointing out this oversight. The numbering a,b,c &d have been gone to the inside.
Question 7: Fig. 6 & 7 should be the same as Fig. 5.
Response 7: Thank you for the useful suggestion provided by the reviewer. Figure 6 and 7 have been revised on Page 15 and 21.
Question 8: The English is also need to edit.
Response 8: Thank you for the useful suggestion provided by the reviewer. The English has been improved and the corresponding changes have been marked in red in the whole manuscript.

Reviewer 2 Report
In my opinion polymers-1819832 manuscript is well written and deserves publication after revision .
Some suggestions to improve the manuscript
1. Check the authors list and move the last author after “and*”.
2. Give abbreviation in the abstract such as SEBS, DA, etc at line 81.
3. Check typos, grammar and topic.
4. At the end of the abstract emphasize the novelty of the paper.
5. At the end of the introduction emphasize originality.
6. Give chemical structure and/or reaction mechanism.
7. Give SEBS yarn EDX data. Also FTIR analysis of all yarns would be very much welcome.
8. At line 142 PDA@n-S... should be PDA@u-S …
9. Figures style is not unitary. For example fig 3 and 8 are too big and fig 4a, 4i too small. So some figures size should decrease and other figures size should increase. Also increase text size of very small figures, axis name and scale are illegible.
1. Compare the properties of the obtained yarns sensor with other sensors from literature.
Author Response
The authors are most grateful for the insightful comments and the helpful recommendations suggested by the reviewers. Once addressed in the final manuscript they will be of undoubted benefit and will improve the paper accordingly. We are thankful for the opportunity to carry out the changes which are highlighted in red text in the revised manuscript.
Each point raised by the reviewers is addressed in italics text in the authors’ response following each point made by the reviewer.
Reviewer #2:
The authors are happy to see the recognition of their work from this reviewer and again are most grateful for their helpful comments and suggestions.
Question 1: Check the authors list and move the last author after “and*”.
Response 1: The authors thank the reviewer for pointing this out. It has been revised on Page 1.
Question 2: Give abbreviation in the abstract such as SEBS, DA, etc at line 81.
Response 2: Thank you for the useful suggestion provided by the reviewer. The full name and abbreviation have been given in the abstract on Page 2.
Question 3: Check typos, grammar and topic.
Response 3: The authors thank the reviewer for pointing this out. The typos, gramma and topic have been checked out.
Question 4: At the end of the abstract emphasize the novelty of the paper.
Response 4: We thank the reviewer for this suggestion. The novelty has been marked in red.
Question 5: At the end of the introduction emphasize originality.
Response 5: The authors thank the reviewer for pointing this out. The originality has been marked in red on Page 5.
Question 6: Give chemical structure and/or reaction mechanism.
Response 6: The authors are grateful for the reviewer pointing out this oversight. The reaction mechanism has been given in Figure 1e on Page 7.
Question 7: Give SEBS yarn EDX data. Also FTIR analysis of all yarns would be very much welcome.
Response 7: Thank you for the useful suggestion provided by the reviewer. The SEBS yarn ESX data was given in Figure 2 on Page 8.
Question 8: At line 142 PDA@n-S... should be PDA@u-S …
Response 8: Thank you for the useful suggestion provided by the reviewer. It was corrected on Page 9.
Question 9: Figures style is not unitary. For example fig 3 and 8 are too big and fig 4a, 4i too small. So some figures size should decrease and other figures size should increase. Also increase text size of very small figures, axis name and scale are illegible.
Response 9: The authors thank the reviewer for pointing this out. The figure style has been checked out.
Question 10: Compare the properties of the obtained yarns sensor with other sensors from literature.
Response 10: Thank you for the useful suggestion provided by the reviewer. There was a comparison on Page 20.
